# Electrochemical Detection of Ammonia in Water Using NiCu Carbonate Hydroxide-Modified Carbon Cloth Electrodes: A Simple Sensing Method

**DOI:** 10.3390/s24154824

**Published:** 2024-07-25

**Authors:** Guangfeng Zhou, Guanda Wang, Xing Zhao, Dong He, Chun Zhao, Hui Suo

**Affiliations:** State Key Laboratory of Integrated Optoelectronics, College of Electronic Science and Engineering, Jilin University, Changchun 130012, China; zhougf22@mails.jlu.edu.cn (G.Z.); gdwang19@mails.jlu.edu.cn (G.W.); zhaoxing1920@mails.jlu.edu.cn (X.Z.); hedong@jlu.edu.cn (D.H.); zchun@jlu.edu.cn (C.Z.)

**Keywords:** ammonia, electrochemical detection, Ni Cu double hydroxide

## Abstract

Excessive ammonia nitrogen can potentially compromise the safety of drinking water. Therefore, developing a rapid and simple detection method for ammonia nitrogen in drinking water is of great importance. Nickel–copper hydroxides exhibit strong catalytic capabilities and are widely applied in ammonia nitrogen oxidation. In this study, a self-supported electrode made of nickel–copper carbonate hydroxide was synthesized on a carbon cloth collector via a straightforward one-step hydrothermal method for rapid ammonia nitrogen detection in water. It exhibits sensitivities of 3.9 μA μM^−1^ cm^−2^ and 3.13 μA μM^−1^ cm^−2^ within linear ranges of 1 μM to 100 μM and 100 μM to 400 μM, respectively, using a simple and rapid i-t method. The detection limit is as low as 0.62 μM, highlighting its excellent anti-interference properties against various anions and cations. The methodology’s simplicity and effectiveness suggest broad applicability in water quality monitoring and environmental protection, particularly due to its significant cost-effectiveness.

## 1. Introduction

In recent years, chloramines, which have low toxicity, have been widely welcomed in the disinfection process. However, improper handling may result in high levels of ammonia nitrogen in tap water [1,2,3]. Ammonia nitrogen has certain physiological toxicity, and excessive levels can cause poisoning in aquatic organisms [4,5]. Additionally, ammonia nitrogen can lead to eutrophication, produce nitrates, consume dissolved oxygen, and severely impact ecosystems and human health [6]. Therefore, the World Health Organization (WHO) recommends that ammonia nitrogen in drinking water should be controlled below 0.5 mg/L (approximately 30 μM) [7]. Thus, the efficient, rapid, and accurate detection and monitoring of ammonia nitrogen concentrations in water are of significant practical importance.

Optical methods, such as Nessler’s reagent spectrophotometry [8], fluorescence analysis [9], and ion chromatography [10], have been widely used for the detection of ammonia nitrogen in water. These methods offer good sensitivity at low concentrations but often require sample pretreatment, making them complex and limiting their real-time monitoring capability. In contrast, potentiometric electrochemical methods, such as gas-sensitive ammonia electrodes [11] and ion-selective electrodes [12,13], are simple to operate, provide rapid response, and have also been widely used for ammonia nitrogen detection. However, these methods are easily affected by various anions and cations, require additional circuit compensation, and are therefore relatively expensive. In contrast, voltametric and amperometric methods, which are based on electrochemical catalysis, detect ammonia nitrogen through redox currents. These methods offer the advantage of resistance to anion and cation interference and have gained considerable attention.

The modification of electrodes with nanomaterials can significantly enhance their catalytic activity and improve sensing sensitivity. For instance, Zhang et al. prepared a Pt/Ni(OH)_2_ electrode via a simple water bath method and achieved ammonia nitrogen detection in the linear range of 5 μM–1000 μM using differential pulse voltammetry (DPV), with a detection limit of 2.3 μM [14]. Wang et al. designed PtCu/CC [15] and PtZn/CC [16] alloy-sensitive electrodes through an alloying strategy and electrodeposition method, improving their corrosion and poisoning resistance. Their sensitivities were 9.5 μA μM^−1^ cm^−2^ and 21.5 μA μM^−1^ cm^−2^, with detection limits of 0.32 nM and 27.81 nM, respectively.

To reduce costs, researchers have also used other non-precious metal materials for ammonia nitrogen detection. For example, Ahmed et al. synthesized ZnO/BiOCl heterojunction materials via a simple hydrothermal method [17], achieving a sensitivity of 11.8 μA μM^−1^ cm^−2^ in the range of 200 Μm–1000 μM using cyclic voltammetry. Kosa et al. prepared Cu_2_O films on glassy carbon electrodes [18] and achieved a sensitivity of 50 μA mM^−1^ cm^−2^ in the range of 10 μM–1000 μM using square wave voltammetry. However, since most of these methods are based on voltammetry, their response speed is relatively slow.

Currently, there have been studies dedicated to using simple and rapid amperometric methods for ammonia nitrogen detection. However, i-t studies targeting ammonia nitrogen detection in water remain scarce. Lu et al. prepared a self-supported Ag/Cu_2_O/TNTs electrode via electrodeposition [19], which was used for i-t detection in a neutral environment, achieving a linear range of 0.1 μM–101 μM and a detection limit as low as 0.074 μM. Wang et al. developed a SPEC/AuNPs/PMB-modified electrode using electrodeposition and electropolymerization [20], indirectly measuring ammonia nitrogen concentration by assessing NADH consumption, with a detection limit of 0.65 μM. Yang et al. synthesized Cu NPs/CC via hydrothermal reduction [21], achieving a linear range of 5–9525 μM and a detection limit of 1.25 μM. However, most existing i-t methods require the use of metal nanomaterials, which are prone to corrosion by chloride and fluoride ions, leading to reduced current. Additionally, the use of precious metals in these electrode materials results in low cost-effectiveness.

Transition metals such as nickel and copper have been proven to be effective materials for ammonia catalysis [22,23]. However, research indicates that the primary active species during ammonia catalysis are their hydroxides. Therefore, directly synthesizing hydroxides is a more efficient approach. Transition metal hydroxides are resistant to corrosion by various anions due to their strong local negative charges [24], and they exhibit good catalytic activity and low cost, making them widely used in supercapacitors [25], water electrolysis [26,27], and other fields. Nickel–copper double hydroxides specifically stand out as highly effective materials for catalyzing ammonia nitrogen oxidation, surpassing even the performance of commercial platinum–carbon electrodes [28,29]. This superior performance stems from the complementary properties of nickel and copper: copper offers a lower onset potential, while nickel provides enhanced adsorption capacity for ammonia nitrogen. Although the optimal element ratio for ammonia nitrogen catalysis has been established at 8:2 [30], different proportions may be optimal in sensor applications due to varying requirements for current and potential in detection contexts. However, hydroxides are prone to aggregation, reducing their electrochemically active surface area and conductivity. An effective strategy is to control their intercalated ions to expand the interlayer spacing, effectively improving their morphology and preventing aggregation. Theoretical calculations by Mohammadi, S et al. have shown that intercalating substances such as water, carbonate, and lactate can enhance the stability of double hydroxides [31]. Yang, A. Z et al. synthesized a tire-like material of nickel–copper double hydroxides intercalated with 1,2-hexanediol (NiCu-DHTs)via a solvothermal method [32], achieving an oxidation current 4.3 times that of commercial platinum–carbon and using it for ammonia nitrogen catalysis and detection. Using a simple i-t method under a continuous addition of 225 μM NH_4_OH, the detection limit was 9 μM. However, like most double hydroxide materials, NiCu-DHTs are bonded to the electrode material with adhesives, resulting in poor reproducibility, stability, and low conductivity, which are not conducive to detection.

In this study, nickel–copper carbonate hydroxides were synthesized on the surface of a carbon cloth current collector via a hydrothermal method, exhibiting high electrochemical activity and relatively fast electron transfer, facilitating the reaction, as schematically illustrated in Figure 1. The insertion of carbonate enhanced its anti-interference properties against various anions and improved its stability. The prepared electrode was tested for ammonia nitrogen sensitivity using a simple and rapid i-t method, showing a wide detection range, high sensitivity, good reproducibility, stability, and excellent anti-interference properties against various anions and cations.

## 2. Materials and Methods

### 2.1. Experimental Instruments and Reagents

Reagents, materials, and apparatus are provided in the Appendix A.

### 2.2. Preparation of Nickel–Copper Hydroxide Electrode

The carbon cloth (CC) was cut into 1.5 cm × 1 cm pieces, and ultrasonically cleaned with toluene, acetone, ethanol, and deionized water for 10 min each three times. The cleaned carbon cloth was then placed in deionized water for later use. Two pieces of pretreated carbon cloth were added to each reactor, and a solution of 30 mL containing 35 mM metal nitrate and 70 mM urea was poured into a hydrothermal reactor, processed at 120 °C for 6 h. Based on the molar ratio of nickel nitrate to copper nitrate (Ni:Cu = 10:0, 8:2, 0:10), the obtained electrode samples were named NC-10@CC, NC-8@CC, and NC-0@CC, respectively.

### 2.3. Electrochemical Measurements

A three-electrode system was used, with the carbon cloth electrode loaded with sensitive materials and a 1 cm × 1 cm platinum sheet electrode serving as the working electrode and counter electrode, respectively. The effective area of the working electrode immersed in the solution was 0.5 cm × 0.5 cm. Prior to electrochemical testing, the electrode was fully activated via 100 cyclic voltammetry scans in the range of 0.2 V–0.7 V in 0.2 M KOH electrolyte, then rinsed with deionized water, and dried. For the electrochemical active area test and electrochemical impedance spectroscopy (EIS), a 50 mL electrolyte containing 5 mM K_3_[Fe(CN)_6_] in 0.1 M KCl was used, with a CV scan rate of 50 mV/s and an EIS frequency range of 0.1 Hz–10^6^ Hz. For ammonia nitrogen sensitivity testing, a 50 mL 0.2 M KOH electrolyte was used.

## 3. Results

### 3.1. Characterization

The morphology of the NC-8@CC electrode and bare carbon cloth was analyzed using scanning electron microscopy (SEM), and the results are shown in Figure 1a–d. Figure 1c,d show the bare CC. Figure 1c is a higher magnification image, illustrating the smooth morphology of the CC surface. Figure 1d is a lower magnification image, further confirming the smooth and clean surface of the CC. In contrast, Figure 1a,b show the NC-8@CC electrode. As seen in Figure 1a, the NC-8 material forms a loose, porous structure composed of nanoneedles [33,34,35] with an edge length of about 500 nm on the carbon cloth surface. Figure 1b shows the NC-8 material completely covering the surface of the carbon cloth, demonstrating that the material successfully grew on the carbon cloth current collector.

The crystal structures of NC-10@CC, NC-8@CC, NC-0@CC, and CC were analyzed via X-ray diffraction (XRD), as shown in Figure 2. All samples exhibit a broad diffraction peak between 20 and 30 degrees, attributed to the graphite phase of the carbon cloth. For the NC-8@CC material, diffraction peaks corresponding to Glaukosphaerite(JCPDS-27-0178) [36] appear at 11.95°, 14.85°, 17.58°, 24.17°, and 31.48°, corresponding to the (110), (020), (120), (220), and (111) crystal planes, respectively. Compared to NC-8@CC, the (110) plane diffraction peak of NC-10@CC is significantly higher, consistent with the characteristics of nickel carbonate hydroxide(JCPDS-35-0501) [37]. In contrast, NC-0 shows more diffraction peaks, which we believe are due to the presence of copper during the hydrothermal process, promoting the formation of a malachite(JCPDS-01-0959) [38] structure in NC-8, thereby exposing more crystal planes beneficial for electrochemical reactions. Unlike typical hydroxides, this carbonate form can reduce agglomeration, enhance material transport and charge transfer, and improve corrosion resistance [39,40].

The chemical composition and elemental ratio of the NC-8@CC electrode were studied in detail via XPS analysis, as shown in Figure 3. In Figure 3a, peaks appear near the binding energies of 288 eV, 531 eV, 855 eV, and 935 eV, corresponding to the binding energies of C 1s, O 1s, Ni 2p, and Cu 2p, respectively. Figure 3b shows that the two peaks at 531.40 eV and 533.0 eV for O1s are attributed to metal carbonate and hydroxide [31]. The fine spectrum of C 1s in Figure 3c can be divided into three peaks at 284.8 eV, 287.0 eV, and 288.3 eV, corresponding to the C-C bond of the carbon cloth, CO_3_^2−^ in (Ni(OH)_2_)_0_._75_(H_2_O)_0_._16_(NiCO3)_0_._09_ [41], and CO_3_^2−^ in malachite [42]. These results indicate the successful synthesis of nickel–copper carbonate hydroxide materials on the carbon cloth surface. In Figure 3d, the peaks at 935.3 eV and 955.1 eV, as well as the satellite peaks at 943.4 eV and 964.2 eV, belong to the 2p3/2 and 2p1/3 of Cu(II). These binding energies are higher than that of copper carbonate hydroxide (934.6 eV) [42], indicating that Cu(II) enters the nickel malachite lattice and is influenced by lattice effects. Similarly, in Figure 3e, Ni shows peaks at 856.3 eV and 874.1 eV, corresponding to 2p3/2 and 2p1/3, and satellite peaks at 863.0 eV and 881.2 eV. These binding energies are also higher than those of nickel carbonate hydroxide (855.8 eV) [43], reflecting the good combination of Ni and Cu.

The elemental ratio of NC-8@CC was found to be 75.15%:24.85%, which is slightly lower than the original feed ratio of 8:2. This may be because, during the hydrothermal process, copper carbonate hydroxide served as a nucleation site, promoting the growth of nickel materials on the carbon cloth surface, resulting in a higher elemental ratio.

### 3.2. Electrochemical Behaviors

To further understand the electrochemical properties of the prepared electrodes, the electrochemical active surface area was estimated using cyclic voltammetry (CV) in a 0.1 M KCl solution containing 5 mM K_3_[Fe(CN)_6_], as shown in Figure 4. Figure 4a exhibits six current peaks in the solution, with peaks I and VI corresponding to the redox process of K_3_[Fe(CN)_6_]. This pair of peaks is observable on all electrodes, with the NC-8@CC electrode showing the highest peak current, according to the formula
*Ipa* = (2.69 × 10^5^) n^3/2^ ACD^1/2^ ν^1/2^(1)

Its electrochemical active surface area (ECSA) can be calculated, where A is the ECSA, n is the number of electrons transferred (in this case, 1), D is the diffusion coefficient of ferricyanide in water (7.60 × 10^−6^ cm^2^ s^−1^), and C is the concentration of the probe molecule (5 × 10^−6^ mol cm^−3^). The ECSA for NC-8@CC, NC-10@CC, and NC-0@CC are 1.66 cm^2^, 1.452 cm^2^, and 1.064 cm^2^ (Appendix A), respectively, with NC-8@CC exhibiting the largest electrochemical active surface area among the modified electrodes. However, it is important to note that the unmodified CC electrode displays an even larger ECSA (2.83 cm^2^). Additionally, the peak potentials of the modified electrodes are significantly shifted compared to the CC electrode, indicating slower charge transfer. This slower charge transfer is likely due to the inherent low conductivity of hydroxides. Nevertheless, this reduction in electrochemical activity is entirely justified by the greatly enhanced ammonia nitrogen sensitivity of the modified electrodes.

Peaks II and V correspond to the complexation of copper species with [Fe(CN)_6_]^3−^, involving a redox process from divalent to monovalent states. Weak peaks are also observed on the NC-0 electrode, indicating that the coexistence of nickel and copper species in the NC-8@CC electrode accelerates charge transfer and enhances the electrochemical activity of the copper species, facilitating detection. Peaks III and IV correspond to the redox process of Ni(II) to Ni(III), which is also observable on the NC-10@CC electrode. Additionally, as shown in Figure 4b, the EIS demonstrates that the NC-8 material has the smallest impedance semicircle radius in modified electrodes, indicating minimal charge transfer resistance.

To evaluate the sensitivity of the electrodes to ammonia nitrogen, their response to 5 mM NH_4_Cl was tested, as shown in Figure 4c,d. Upon the addition of ammonia nitrogen, an enhanced current at 0.65 V–0.7 V is observed for the NC-8 material, whereas other electrodes show almost no response. For a deeper understanding of the electrochemical process of ammonia nitrogen on the surface of the NC-8@CC electrode, tests were conducted at a low scan rate of 1 mV/s. Figure 5a shows a peak at 0.55 V corresponding to the transition from Ni(II) to Ni(III), while the catalytic peak potential of ammonia nitrogen appears around 0.6 V. The sharp current increase near 0.7 V is attributed to the oxygen evolution reaction. The absence of a corresponding reduction peak for ammonia nitrogen oxidation indicates an irreversible reaction. A further analysis of the changes in peak potential and peak current at different scan rates is presented in Figure 5b–d. With an increasing logarithm of the scan rate, the peak potential increases linearly; while with an increasing square root of the scan rate, the peak current increases linearly, indicating a diffusion-controlled process. According to Laviron’s theory [44] for irreversible processes, the peak anodic potential (*E_pa_*) and scan rate (*v*) are given by the following equation: (2)$$E_{pa}=E_{0}-\frac{RT}{(1-\alpha )nF}ln(\frac{RTk}{(1-\alpha )nF})+\frac{RT}{(1-\alpha )nF}ln(v)$$
where $E_{0}$ is the formal oxidation/reduction potential, *k* is the standard rate constant of the electrochemical reaction (typically corresponding to the reduction reaction for the cathode and oxidation reaction for the anode), *n* is the number of electrons transferred in the reaction, and α is the charge transfer coefficient, usually taken as 0.5. *R*, *T*, and *F* are the universal gas constant (8.31 J·mol^−1^·K^−1^), the absolute temperature (298 K), and the Faraday constant (96,485.33 C·mol^−1^), respectively. The final calculated number of electrons transferred is 1.38. By consulting the standard potential table, we propose that the possible final product is NH_3_OH^+^ [45], following the following reaction process:NH_3_ + OH^−^ − e^−^ = NH_2_* + H_2_O(3)
2NH_2_* = N_2_H_4_
(4)
N_2_H_4_ + H_2_O ⇌ N_2_H_5_^+^ + OH^−^
(5)
N_2_H_5_^+^ + 2H_2_O − 2e^−^ ⇌ 2NH_3_OH^+^ + H^+^ (1.42 V vs. NHE.)(6)

In the typical ammonia oxidation process, the coupling of NH_2_ is the rate-limiting step [46]. In considering that reactions (5) and (6) facilitate reaction (3), this could be the reason for the strong oxidation peak observed at 0.60 V. Reaction (5) is difficult to conduct in an alkaline environment, resulting in an incomplete reaction, which might explain why the number of electrons transferred is between 1 and 2. In summary, the NC-8@CC electrode exhibits a high response to ammonia nitrogen, primarily due to the strong catalytic oxidation capability of the material itself toward ammonia nitrogen. 

### 3.3. Sensitive Determination of Ammonia

In using chronoamperometry, the linear detection of ammonia nitrogen by the NC-8@CC electrode was investigated. To determine the optimal detection potential, responses to continuously added NH_4_Cl were tested at 0.50 V, 0.55 V, 0.60 V, and 0.65 V, as shown in Figure 6a. At 0.50 V, there was no response to the continuous addition of NH_4_Cl, indicating that the reaction had not initiated. A weak step response was observed at 0.55 V. At both 0.60 V and 0.65 V, a strong step response was observed, although the current at 0.65 V was slightly lower compared to that at 0.60 V. In considering energy consumption and interference resistance, 0.60 V was chosen as the detection potential for ammonia nitrogen.

Under 0.60 V, the NC-8@CC electrode exhibited rapid increases in current response with successive additions of ammonia nitrogen, displaying a stair-step pattern, as shown in Figure 6b. According to Figure 6c, excellent linear relationships between the current response and concentration are observed over large ranges of 1 μM–100 μM and 100 μM–400 μM, with equations y = 3.9 × 10^−3^ × (mAμM^−1^ cm^−2^) + 0.1336 (R^2^ = 0.999, 5–100 μM) and y = 3.13 × 10^−3^ × (mA μM^−1^ cm^−2^) + 0.2288 (R^2^ = 0.996, 100–400 μM), respectively. The detection limit was as low as 0.62 μM (S/N = 3), significantly lower than the WHO’s recommended limit of 30 μM for drinking water. Compared to recent reports on electrochemical sensors (Table 1), NC-8@CC demonstrates advantages such as a wide linear range, high sensitivity, low detection limit, and rapid response. While its sensitivity may not match that of precious metal-based materials, it achieves a practical level of performance at a significantly lower cost. This makes NC-8@CC a more cost-effective solution for ammonia nitrogen detection.

Additionally, compared to our previous platinum-based materials, which require methods such as cyclic voltammetry (CV) and differential pulse voltammetry (DPV), which involve stirring, settling, and slow scanning, the i-t method used in this study allows for real-time response and faster detection. This makes the NC-8@CC electrode not only more affordable but also more efficient in practical applications.

### 3.4. Anti-Interference, Stability, Repeatability, and Real-Sample Analysis

The interference resistance of the NC-8@CC electrode was tested to evaluate its potential for practical applications, as shown in Figure 7a. Interfering substances such as NaCl, KCl, NaF, Na_2_CO_3_, Na_2_SO_4_, and NaNO_3_ were considered, while calcium and magnesium ions were excluded as they precipitate at this pH. After adding interfering substances at 20 times the concentration of the analyte, the electrode’s current response remained stable without significant changes. According to Figure 7b, the calculated RSD is only 0.43%, demonstrating its good resistance to common interfering ions. Notably, fluoride ions, which are highly corrosive and typically cause significant response current decreases in alloy-based and glass electrodes, had minimal impact. The NC-8@CC carbon cloth electrode, utilizing a carbonate hydroxide as the sensitive material, is not sensitive to fluoride ions and thus remains stable in fluoride-containing environments such as tap water and seawater.

Figure 7c shows the i-t response of the NC-8@CC electrode to 50 μM NH_4_Cl across five repetitions, with an RSD of 1.82%. Figure 7d indicates that the current response of the NC-8@CC electrode changed minimally over 10 days, with an RSD of only 3.73%, reflecting good stability. This stability may be attributed to the introduction of carbonate ions in the NC-8 material, which enhances local negative charge, repels various anions, and improves overall material stability.

The samples of tap water were collected as the actual test samples. The practical application value of NC-8@CC electrodes was evaluated using the calibration method. The results are shown in Table 2. The recovery ranged from 102.5 to 105.9%, and the relative standard deviation (RSD) was less than 3.3%. The NC-8@CC electrode demonstrated good performance in water with a low ammonia concentration and showed potential for practical application.

## 4. Conclusions

In summary, a novel self-supporting electrode was successfully synthesized through hydrothermal methods, producing nickel–copper carbonate hydroxide material on a carbon cloth current collector. This electrode demonstrates the rapid and highly sensitive detection of ammonia nitrogen in water, characterized by excellent catalytic activity and reasonable charge transfer properties, along with good stability. It exhibits high sensitivity with 3.9 μA μM^−1^ cm^−2^ in the range of 1 μM to 100 μM and 3.13 μA μM^−1^ cm^−2^ in the range of 100 μM to 400 μM, with a detection limit as low as 0.62 μM. Although its sensitivity is not as high as that of precious metal-based materials, it achieves a practical level of performance at a much lower cost. Additionally, when tested in tap water, it showed good recovery rates, further demonstrating its potential for real-world applications. Its ability to maintain stability and sensitivity even in the presence of common interfering ions indicates its potential utility in real-world water quality assessment.

## Data Availability

Data will be made available upon request.

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
