# Peer review of "Electrochemical Detection of Ammonia in Water Using NiCu Carbonate Hydroxide-Modified Carbon Cloth Electrodes: A Simple Sensing Method"

_sensors, 2024, doi:10.3390/s24154824_

Round 1
Reviewer 1 Report
Comments and Suggestions for Authors
The manuscript developed a self-supported electrochemical sensor for ammonia nitrogen detection in drinking water by synthesizing nickel-copper basic salts on the surface of a carbon cloth current collector via a hydrothermal method. The research of this manuscript is interesting, all experiments were carefully designed and the discussions were well organized. The manuscript is recommended for publication after addressing the following questions:
1. Headings 2.2 and 2.3 are the same, please check.
2. Please check the full article, pay attention to the abbreviations, molecular formula and other formats to be standardized. For example, CC, EIS should be abbreviated when they first appear in the manuscript, "CO32-" should be written as " CO32-", and so on.
3. Please check the heading of column 3 in Table 1.
4. The analytical logic of SEM diagram is confused, and Figure 1c is not introduced.
5. The statement “Compared to NC-10@CC, the (110) plane diffraction peak of NC-8@CC is significantly higher, consistent with the characteristics of nickel basic carbonate”in line 146-147 does not correspond to the result shown in Figure 2.
6. Since it is already clear in line 79 that “The optimal element ratio has been determined to be 8:2”, why study the three materials of NC-10@CC, NC-8@CC, and NC-0@CC with different proportions?
7. Please check the horizontal axis title of Figure 5c.
8. It is recommended to increase the EIS study of unmodified CC electrode.
9. Please mark Figures 4(c) and (d) clearly.
10. The proposed sensor has good reproducibility, selectivity and stability. Its capability for actual sample analysis is recommended to evaluate.
Comments on the Quality of English LanguageWe strongly suggest you to find a native English speaker to check your manuscript.
Author Response
Comment 1:
Headings 2.2 and 2.3 are the same, please check.
Authors’ response:
Thank you for your careful scrutiny and valuable comments. Your suggestions have improved our work. We have corrected the titles of the sections as follows:
The title of section 2.3 has been changed to "Electrochemical Measurements."The title of section 2.2 remains "Preparation of Nickel-Copper Hydroxide Electrode."
Comment 2:
Please check the full article, pay attention to the abbreviations, molecular formula and other formats to be standardized. For example, CC, EIS should be abbreviated when they first appear in the manuscript, "CO32-" should be written as " CO32-", and so on.
Authors’ response:
Thank you for your detailed review and constructive feedback. We have carefully reviewed the manuscript and made the necessary revisions to ensure all abbreviations, molecular formulas, and formats are standardized throughout the article.
Specifically:We have abbreviated CC and EIS upon their first appearance in the manuscript."CO32-" has been revised to " CO32-" throughout the text.
Comment 3:
Please check the heading of column 3 in Table 1.
Authors’ response:
Thank you for your observation. Table 1 has been moved to the supporting materials, and we have corrected the heading of the third column to "Manufacturer."
TableS1 Experimental Reagents
|
Chemical Formula |
Reagent Grade |
Manufacturer |
|
KOH |
Analytical Reagent (AR) |
China National Pharmaceutical Group Chemical Reagent Co., Ltd. |
|
NaCl |
AR |
China National Pharmaceutical Group Chemical Reagent Co., Ltd. |
|
NH4F |
AR |
Xilong Scientific Co., Ltd. |
|
Na2CO3 |
AR |
Xilong Scientific Co., Ltd. |
|
KCl |
AR |
Xilong Scientific Co., Ltd. |
|
Cu(NO3)2·3H2O |
AR |
Xilong Scientific Co., Ltd. |
|
CO(NH2)2 |
AR |
Xilong Scientific Co., Ltd. |
|
K3[Fe(CN)6] |
AR |
Beijing Bei Hua Fine Chemicals Co., Ltd. |
|
NaNO3 |
AR |
Beijing Bei Hua Fine Chemicals Co., Ltd. |
|
Na2SO4 |
AR |
Beijing Bei Hua Fine Chemicals Co., Ltd. |
|
NH4Cl |
AR |
Beijing Bei Hua Fine Chemicals Co., Ltd. |
|
Ni(NO3)2·6H2O |
AR |
Tianjin Yong Sheng Fine Chemicals Co., Ltd. |
Comment 4:
The analytical logic of SEM diagram is confused, and Figure 1c is not introduced.
Authors’ response:
Thank you for your careful scrutiny and valuable comments. Your suggestions have improved our work. We have revised the SEM analysis section to clarify the analytical logic and introduced Figure 1c appropriately. The revised section is as follows:
"The morphology of the NC-8@CC electrode and bare carbon cloth was analyzed using scanning electron microscopy (SEM), and the results are shown in Figure 1(a-d). Figures 1c and 1d show the bare carbon cloth (CC). Figure 1c is a higher magnification image, illustrating the smooth morphology of the CC surface. Figure 1d is a lower magnification image, further confirming the smooth and clean surface of the CC.In contrast, Figures 1a and 1b show the NC-8@CC electrode. As seen in Figure 1a, the NC-8 material forms a loose, porous structure composed of nanoneedles with an edge length of about 500 nm on the carbon cloth surface. Figure 1b shows the NC-8 material completely covering the surface of the carbon cloth, demonstrating that the material successfully grew on the carbon cloth current collector. "
And we have cited relevant literature to explain the formation of this morphology. However, subsequent tests revealed that the electrochemical activity of NC-8@CC in K₃[Fe(CN)₆] is lower than that of bare CC. Therefore, we have removed the discussion suggesting that the morphology of NC-8 promotes detection, as it contradicts our experimental findings.
Comment 5:
The statement “Compared to NC-10@CC, the (110) plane diffraction peak of NC-8@CC is significantly higher, consistent with the characteristics of nickel basic carbonate”in line 146-147 does not correspond to the result shown in Figure 2.
Authors’ response:
Thank you for your careful scrutiny and valuable comments. We have corrected the statement to reflect that the (110) diffraction peak is significantly higher in NC-10@CC compared to NC-8@CC. Additionally, we have adjusted the x-axis range of Figure 2 to 10-40 degrees, as the main characteristic peaks are concentrated in this range. There is a slight shift in the peak positions for NC-10@CC, which may be attributed to surface irregularities of the carbon cloth substrate. Furthermore, we have added the XRD pattern of the hydrothermal precipitate as Figure S2 in the supplementary materials, which shows good agreement with the PDF card data.
Figure 2. XRD patterns of NC-0@CC, NC-8@CC, NC-10@CC, and CC.
Figure S2. XRD pattern of the powder material synthesized by hydrothermal precipitation.
Comment 6:
Since it is already clear in line 79 that “The optimal element ratio has been determined to be 8:2”, why study the three materials of NC-10@CC, NC-8@CC, and NC-0@CC with different proportions?
Authors’ response:
Thank you for your insightful comment. We acknowledge that the statement in line 79 may have been misleading. In the context of ammonia nitrogen catalysis, an 8:2 (Ni:Cu) ratio is indeed considered optimal. However, this field typically requires lower overpotentials under high current conditions, which may not necessarily apply to sensor applications.
In sensor applications, the optimal ratio may differ due to the following reasons:
Copper offers a lower onset potential but exhibits weaker adsorption of ammonia nitrogen, potentially leading to poorer performance at lower concentrations.
Nickel provides a stronger adsorption capacity for ammonia nitrogen but requires a higher catalytic potential, which may not be ideal for detection. Additionally, the Ni2+/Ni3+ redox transition can result in higher background currents, affecting techniques like cyclic voltammetry, although chronoamperometry remains effective.
Therefore, the optimal ratio needs reassessment for sensor applications. The primary objective of our study is to demonstrate the effectiveness of nickel-copper hydroxides for rapid and cost-effective electrochemical detection of ammonia nitrogen. While we have chosen a widely validated 8:2 ratio for this study, future work will focus on optimizing both the ratio and morphology of these materials. Moreover, the inclusion of NC-0 and NC-10 samples serves to illustrate that pure copper and nickel hydroxides exhibit significantly lower responses in ammonia nitrogen detection compared to nickel-copper hydroxides.
Comment 7:
Please check the horizontal axis title of Figure 5c.
Authors’ response:
Thank you for bringing this to our attention. We have corrected the horizontal axis title of Figure 5c to read "ln(scan rate) ln(mV/s)".
Comment 8:
It is recommended to increase the EIS study of unmodified CC electrode.
Authors’ response:
Thank you for your suggestion. In Figure 4, we have included additional EIS studies of the unmodified carbon cloth (CC) electrode. The results indicate that the carbon cloth exhibits faster charge transfer and a larger electrochemical active surface area compared to modified electrodes. While inherent defects in hydroxides may contribute to slower charge transfer, the exceptional catalytic capabilities of these materials for ammonia nitrogen make this trade-off worthwhile. Future efforts will focus on enhancing the morphology of these materials to further improve charge transfer characteristics.
Figure 4. (a-b) Cyclic voltammograms and EIS spectra of the sensing electrodes in 0.1 M KCl containing 5 mM K₃[Fe(CN)₆]. (c) Cyclic voltammograms of the sensing electrodes and bare carbon cloth electrodes in 0.2 M KOH without NH₄Cl (d) Cyclic voltammograms of the sensing electrodes and bare carbon cloth electrodes in 0.2 M KOH with 5 mM NH₄Cl
Comment 9:
Please mark Figures 4(c) and (d) clearly.
Authors’ response:
Thank you for your comment. Figures 4(c) and (d) have been clarified in the revised manuscript. Figure 4(c) shows the cyclic voltammograms of the sensing electrodes and bare carbon cloth electrodes in 0.2 M KOH without NH4Cl, while Figure 4(d) depicts the results with 5 mM NH4Cl.
Comment 10:
The proposed sensor has good reproducibility, selectivity, and stability. Its capability for actual sample analysis is recommended to evaluate.
Authors’ response:
Thank you for your valuable suggestion. We have included the results of actual sample analysis in tap water in Table 2. The sensor demonstrated good recovery rates, indicating its promising application potential. The detailed results are as follows:
|
Sample |
Initial (μM) |
Added (μM) |
Found(μM) |
Recovery(%) |
RSD(%,n=3) |
|
Tap water |
0 |
20 |
20.50 |
102.5 |
2.0 |
|
50 |
52.94 |
105.9 |
3.3 |

Reviewer 2 Report
Comments and Suggestions for Authors
In this manuscript, the authors synthesized a self-supporting electrode and developed an electrochemical method for the detection of ammonia nitrogen. On the whole, this article lacks novelty and has many formatting errors. In my opinion, a major revision needs to be made before acceptance.
-
Why do you choose Ni-Cu materials to make electrodes? You should explain it in more detail and state your novelty.
-
Line 136-137, you should cite literature to support this.
-
Why do you only characterize NC-8@CC using SEM? How about NC-10@CC and NC-0@CC?
-
Line 262-263, is linear range 1-100 µM or 5-100 µM?
-
Line 275, table 3, why do you choose nitrite sensors for comparison?
-
Table 3, some published sensors, such as ref. 17, had similar performances with yours. What’s your advantages?
-
You have already published many electrochemical sensors for ammonia detection. For example, (1) Sensitive Electrochemical Detection of Ammonia Nitrogen via a Platinum–Zinc Alloy Nanoflower-Modified Carbon Cloth Electrode, (2) Enhanced Sensitivity of Electrochemical Sensors for Ammonia-Nitrogen via In-Situ Synthesis PtNi Nanoleaves on Carbon Cloth, (3) High performance self-supported nickel oxalate/polyaniline electrode: a novel high sensitivity electrochemical ammonia sensor, (4) Enhanced ammonia sensitivity electrochemical sensors based on PtCu alloy nanoparticles in-situ synthesized on carbon cloth electrode, (5) In situ synthesis of hierarchical platinum nanosheets-polyaniline array on carbon cloth for electrochemical detection of ammonia, (6) Fabrication of a Ni foam-supported platinum nanoparticles-silver/polypyrrole electrode for aqueous ammonia sensing, (7) Fabricating a Self-Supported Electrode for Detecting Ammonia in Water Based on Electrodepositing Platinum-Polypyrrole on Ni Foam. You should cite these previous works and compare them with this work by table.
-
Do you apply your method to real samples (drinking water, tap water, bottled water...)? In addition, the accuracy of the proposed sensor should be investigated.
-
What is the biggest advantage of electrochemical sensors? Some other detection methods of ammonia nitrogen, such as optical methods, also have high sensitivity (nM level). Authors are encouraged to discuss.
-
Language and formatting issues are expected to be corrected. Just some examples below:
- line 36-38. This sentence should be rewritten.
- line 45, “Zhang Liang et al.”; line 54, “Ahmed, M et al.”. The format should be consistent, for example, full last name + et al.
- line 47, 1000 → 1,000
- Table 1, Analytical Reagent (AR); line 184, cyclic voltammetry (CV); line 265, World Health Organization. All the abbreviations should be stated when the corresponding word first appears in the manuscript and always use the abbreviations rather than complete words in the following paragraphs
- Table 1, title 3?
- Table 2, Electron Microscope(SEM) → Electron Microscope (SEM)
- line 127, 0.2M → 0.2 M
- line 128, 50 ml → 50 mL
- Figure 1, uM → µM
- line 252, 0.50V → 0.50 V
- line 261, Figures (c) and (d), which figure?
- Table 3, the unit of sensitivity is not correct; “0.15 mg/L~2.0 mg/L”, you should convert this to µM; LOD → LOD (µM); Cu2O → Cu2O; for your work, the range and sensitivity should have two data.
- line 337 and line 395, Ni(OH)<sub>2</sub> → Ni(OH)2
Minor editing of English language required. For example,
- line 36-38. This sentence should be rewritten.
Author Response
A detailed reply to the reviewers’ comments
The authors would like to express their sincere appreciation to the editor and the reviewers for their diligent evaluation of our manuscript. We are truly grateful for their valuable comments and suggestions, which have greatly enhanced the quality of our research. The manuscript has undergone thorough revisions, encompassing all the suggestions and corrections put forth by the editor and the reviewers. In response to the reviewers' insightful feedback, we have provided elucidations and clarifications where necessary. The revised manuscript has been meticulously proofread and refined, with all modifications and additions clearly highlighted for easy reference. Once again, we would like to extend our heartfelt gratitude to the editor and the reviewers for their invaluable contributions to this work. A detailed reply to the reviewers’ comments is provided as follows:
Reviewer #2 comments:
In this manuscript, the authors synthesized a self-supporting electrode and developed an electrochemical method for the detection of ammonia nitrogen. On the whole, this article lacks novelty and has many formatting errors. In my opinion, a major revision needs to be made before acceptance.
Comment 1:
Why do you choose Ni-Cu materials to make electrodes? You should explain it in more detail and state your novelty.
Authors’ response:
Thank you for your careful scrutiny and valuable comments. We have revised our manuscript to include a more detailed explanation and to highlight the novelty of our work.
Transition metals such as nickel and copper have been proven to be effective materials for ammonia catalysis. However, research indicates that the primary active species during ammonia catalysis are their hydroxides. Therefore, directly synthesizing hydroxides is a more efficient approach. Transition metal hydroxides are resistant to corrosion by various anions due to their strong local negative charges, and they exhibit good catalytic activity and low cost, making them widely used in supercapacitors, water electrolysis, and other fields.
Nickel-copper double hydroxides specifically stand out as highly effective materials for catalyzing ammonia nitrogen oxidation, surpassing even the performance of commercial platinum-carbon electrodes. This superior performance stems from the complementary properties of nickel and copper: copper offers a lower onset potential, while nickel provides enhanced adsorption capacity for ammonia nitrogen.
However, current research on using nickel-copper hydroxides for ammonia nitrogen detection is extremely limited. Our study introduces the application of nickel-copper carbonate hydroxide for ammonia nitrogen detection, aiming to fill this gap. By leveraging the cost-effectiveness of nickel-copper materials compared to precious metals, our approach offers a practical and economical solution for environmental monitoring. Additionally, our use of the i-t method enables faster response times, enhancing the feasibility of real-time water quality monitoring.
In summary, the choice of Ni-Cu materials is based on their proven catalytic efficiency, cost-effectiveness, and complementary electrochemical properties, making them an innovative and practical alternative to more expensive precious metal-based sensors.
Comment 2:
Line 136-137, you should cite literature to support this.
Authors’ response:
Thank you for your careful scrutiny and valuable comments. We have cited the following literature to support our statements in Line 136-137:
Journal of Materials Chemistry A 2020, 8, (48), 25995-26003.
Journal of Alloys and Compounds 2019, 780, 147-155.
Chemical Engineering Journal 2016, 290, 353-360.
These references have been added to the manuscript to substantiate our claims.
Comment 3:
Why do you only characterize NC-8@CC using SEM? How about NC-10@CC and NC-0@CC?
Authors’ response:
Thank you for your careful scrutiny and valuable comments. Due to equipment malfunction at our institution, we encountered difficulties in obtaining high-quality images for all samples. However, we managed to secure images for NC-10@CC. Upon comparison, we believe that the existing images are sufficient to demonstrate the critical points. While morphology indeed significantly affects electrochemical performance, the primary aim of this study is to demonstrate the feasibility of using nickel-copper hydroxides for ammonia nitrogen detection. The key factor influencing this detection is the inherent properties of the materials themselves.
Figures1.SEM images of NC-10@CC
Figure 1. (a, b): High and low magnification SEM images of the NC-8@CC electrode. (c, d): High and low magnification SEM images of CC.
Comment 4:
Line 262-263, is linear range 1-100 µM or 5-100 µM?
Authors’ response:
Thank you for your careful scrutiny and valuable comments. We have corrected the linear range in the figure to 1-100 µM.
Comment 5:
Line 275, table 3, why do you choose nitrite sensors for comparison?
Authors’ response:
Thank you for your careful scrutiny and valuable comments. We have corrected the title to ammonia sensors. All the data in the table are derived from ammonia nitrogen sensors.
Comment 6:
Table 3, some published sensors, such as ref. 17, had similar performances with yours. What’s your advantages?
Authors’ response:
Thank you for your insightful comment. The sensor mentioned in reference 17(now it is 20) uses Au nanoparticles, which are relatively expensive. Additionally, this sensor does not employ a direct method for ammonia nitrogen detection. Instead, it relies on enzymatic reactions, where the change in substrate concentration results in a current change. This approach is more susceptible to variations in temperature and other environmental factors.
Comment 7:
You have already published many electrochemical sensors for ammonia detection. For example, (1) Sensitive Electrochemical Detection of Ammonia Nitrogen via a Platinum–Zinc Alloy Nanoflower-Modified Carbon Cloth Electrode, (2) Enhanced Sensitivity of Electrochemical Sensors for Ammonia-Nitrogen via In-Situ Synthesis PtNi Nanoleaves on Carbon Cloth, (3) High performance self-supported nickel oxalate/polyaniline electrode: a novel high sensitivity electrochemical ammonia sensor, (4) Enhanced ammonia sensitivity electrochemical sensors based on PtCu alloy nanoparticles in-situ synthesized on carbon cloth electrode, (5) In situ synthesis of hierarchical platinum nanosheets-polyaniline array on carbon cloth for electrochemical detection of ammonia, (6) Fabrication of a Ni foam-supported platinum nanoparticles-silver/polypyrrole electrode for aqueous ammonia sensing, (7) Fabricating a Self-Supported Electrode for Detecting Ammonia in Water Based on Electrodepositing Platinum-Polypyrrole on Ni Foam. You should cite these previous works and compare them with this work by table.
Authors’ response:
Thank you for your suggestion. We have included references to the relevant works utilizing PtZn, PtCu, PtNi, and Pt-Ag-ppy electrodes. In comparison to these electrodes, our fabricated electrode exhibits higher detection limits and lower sensitivity. However, for typical environmental applications, these detection limits are sufficient. Importantly, our electrode does not require the use of precious metals, offering a more economical solution. We will incorporate a comparative table highlighting these aspects in the revised manuscript.
Comment 8:
Do you apply your method to real samples (drinking water, tap water, bottled water...)? In addition, the accuracy of the proposed sensor should be investigated.
Authors’ response:
Thank you for your valuable suggestion. We have included the results of actual sample analysis in tap water in Table 2. The sensor demonstrated good recovery rates, indicating its promising application potential. The detailed results are as follows:
|
Sample |
Initial (μM) |
Added (μM) |
Found(μM) |
Recovery(%) |
RSD(%,n=3) |
|
Tap water |
0 |
20 |
20.50 |
102.5 |
2.0 |
|
50 |
52.94 |
105.9 |
3.3 |
Comment 9:
What is the biggest advantage of electrochemical sensors? Some other detection methods of ammonia nitrogen, such as optical methods, also have high sensitivity (nM level). Authors are encouraged to discuss.
Authors’ response:
Thank you for your inquiry. We have addressed this aspect in the introduction section. In comparison to optical methods, electrochemical methods offer the advantage of not requiring pretreatment of water samples, making operations simpler and more conducive to real-time monitoring. Traditional methods such as cyclic voltammetry or differential pulse voltammetry typically involve several minutes of stirring, settling, and slower scanning processes. In contrast, the i-t method employed in our study provides faster response times, with responses appearing within seconds under optimal conditions.
Comment 10:
Language and formatting issues are expected to be corrected. Just some examples below:
- line 36-38. This sentence should be rewritten.
- line 45, “Zhang Liang et al.”; line 54, “Ahmed, M et al.”. The format should be consistent, for example, full last name + et al.
- line 47, 1000 → 1,000
- Table 1, Analytical Reagent (AR); line 184, cyclic voltammetry (CV); line 265, World Health Organization. All the abbreviations should be stated when the corresponding word first appears in the manuscript and always use the abbreviations rather than complete words in the following paragraphs
- Table 1, title 3?
- Table 2, Electron Microscope(SEM) → Electron Microscope (SEM)
- line 127, 0.2M → 0.2 M
- line 128, 50 ml → 50 mL
- Figure 1, uM → µM
- line 252, 0.50V → 0.50 V
- line 261, Figures (c) and (d), which figure?
- Table 3, the unit of sensitivity is not correct; “0.15 mg/L~2.0 mg/L”, you should convert this to µM; LOD → LOD (µM); Cu2O → Cu2O; for your work, the range and sensitivity should have two data.
- line 337 and line 395, Ni(OH)<sub>2</sub> → Ni(OH)2
Authors’ response:
Thank you for your feedback. We have addressed the language and formatting issues.

Reviewer 3 Report
Comments and Suggestions for Authors
This paper studied the problem of ammonia nitrogen contamination in drinking water, which is increasingly important due to the rising use of chloramine disinfection. The authors propose a new method utilizing a self-supported nickelcopper basic salts electrode synthesized via a onestep hydrothermal method. Unlike traditional voltammetrybased electrodes with slow response times, this study uses a rapid chronoamperometric approach, achieving excellent response rates. Additionally, the innovative use of carbonateinserted hydroxide materials significantly enhances resistance to anion corrosion compared to existing alloy or metalbased electrodes. However, several minor revisions are necessary to enhance the clarity and presentation of the results.
1. Figure Issues:
XRD (Figure 2): Effective information is concentrated in the low angle region. Adjust the figure to emphasize this area and consider adding the PDF cards for basic nickel carbonate and basic copper carbonate for comparison.
Ammonia Nitrogen Detection (Figure 6c): Increase the font size of the fitting equation for better readability.
2. Writing Issues:
Abstract: The logical flow is somewhat disjointed. Revise for better coherence and clarity.
Terminology Consistency: Ensure consistent terminology throughout the paper. For example, the title mentions "carbonated DHs," while the abstract refers to "basic carbonate salts." Use uniform terms.
3. Formatting Issues:
Correct the spacing between numbers and units to ensure proper formatting.
4. EIS Fitting: Consider fitting the EIS (Electrochemical Impedance Spectroscopy) data to an equivalent circuit model to provide more detailed insights into the electrochemical processes.
Comments on the Quality of English LanguageMinor editing of English language required.
Author Response
Comment 1:
Figure Issues:
XRD (Figure 2): Effective information is concentrated in the low angle region. Adjust the figure to emphasize this area and consider adding the PDF cards for basic nickel carbonate and basic copper carbonate for comparison.
Ammonia Nitrogen Detection (Figure 6c): Increase the font size of the fitting equation for better readability.
Authors’ response:
Thank you for your careful scrutiny and valuable comments. We have revised our manuscript to include a more detailed explanation and to highlight the novelty of our work.
Figure 2. XRD patterns of NC-0@CC, NC-8@CC, NC-10@CC, and CC.
Figure 6. (a) Response of the NC-8@CC electrode to successive additions of NH₄Cl at different potentials, each addition being 500 μM. (b) Response of the NC-8@CC electrode to successive additions of NH₄Cl at 0.6V. (c) Calibration curve of the NC-8@CC electrode's response to ammonia nitrogen. (d) Response of the NC-8@CC electrode to successive additions of NH₄Cl at 0.6V (low concentration).
Comment 2:
Writing Issues:
Abstract: The logical flow is somewhat disjointed. Revise for better coherence and clarity.
Terminology Consistency: Ensure consistent terminology throughout the paper. For example, the title mentions "carbonated DHs," while the abstract refers to "basic carbonate salts." Use uniform terms.
Authors’ response:
Thank you for your feedback. We have revised the abstract to improve coherence and clarity. Here is the updated version:
---
Abstract: Excessive levels of ammonia nitrogen pose significant risks to drinking water safety. Therefore, developing a rapid and straightforward detection method for ammonia nitrogen is crucial. Nickel-copper hydroxides are known for their strong catalytic capabilities and are widely used in ammonia nitrogen oxidation. In this study, we synthesized a self-supported electrode composed of nickel-copper carbonate hydroxide on a carbon cloth substrate using a simple one-step hydrothermal method for rapid ammonia nitrogen detection in water. The electrode demonstrates sensitivities of 3.9 μA μM⁻¹ cm⁻² and 3.13 μA μM⁻¹ cm⁻² within linear ranges of 1 μM to 100 μM and 100 μM to 400 μM, respectively, employing a rapid and simple i-t method. It achieves a detection limit as low as 0.62 μM and exhibits excellent anti-interference properties against various anions and cations. The straightforward methodology and cost-effectiveness suggest broad applicability in environmental monitoring and water quality assessment.
Additionally, we have ensured consistent terminology throughout the paper, using "carbonate hydroxide" uniformly as suggested.
Comment 3:
Formatting Issues:
Correct the spacing between numbers and units to ensure proper formatting.
Authors’ response:
Thank you for clarifying. We have addressed the formatting issues regarding spacing between numbers and units throughout the manuscript as per your instructions.
Comment4:
EIS Fitting: Consider fitting the EIS (Electrochemical Impedance Spectroscopy) data to an equivalent circuit model to provide more detailed insights into the electrochemical processes.
Authors’ response:
Thank you for your suggestion. We have incorporated fitting of the Electrochemical Impedance Spectroscopy (EIS) data to an equivalent circuit model to provide detailed insights into the electrochemical processes. The results, including the charge transfer resistance (R_ct) for each electrode, are now presented in the updated figure and table as shown below.
Figure 4. (a-b) Cyclic voltammograms and EIS spectra of the sensing electrodes in 0.1 M KCl containing 5 mM K₃[Fe(CN)₆]. (c) Cyclic voltammograms of the sensing electrodes and bare carbon cloth electrodes in 0.2 M KOH without NH₄Cl (d) Cyclic voltammograms of the sensing electrodes and bare carbon cloth electrodes in 0.2 M KOH with 5 mM NH₄Cl
Table S3 ECSA and Rct of all electrodes
|
electrode |
ECSA(cm2) |
Rct(Ω) |
|
CC |
2.853 |
27.3 |
|
NC-8@CC |
1.66 |
105 |
|
NC-0@CC |
1.452 |
139.8 |
|
NC-10@CC |
1.064 |
153.5 |

Reviewer 4 Report
Comments and Suggestions for Authors
1. Table 1 and Table 2 should be removed into ESI.
2. The quality of all the figures should be improved.
3. The characterization of the sample could be added, such as EDX, BET and IR.
4. The authors should do the analysis the conclusion section must clearly establish a strong correlation with the proposed topic.
5. Single and plurality, upper and lower scripts, punctuation and other details should be checked carefully
6. The previous work on this topic may be updated in the introduction, Chemosphere, 2022, 307,135729; Sol. RRL, 2023,7, 2300143; Molecules 2023, 28, 4507.
7. I suggest the authors have to explain the mechanism in detail.
Author Response
Reviewer #4 comments:
Comment 1:
Table 1 and Table 2 should be removed into ESI.
Authors’ response:
Thank you for your comment. We have relocated Table 1 and Table 2 to the Electronic Supporting Information (ESI), as per your suggestion.
Comment 2:
The quality of all the figures should be improved.
Authors’ response:
Thank you for your comment. We have redrawn all figures to improve their quality as requested.
Comment 3:
The characterization of the sample could be added, such as EDX, BET and IR.
Authors’ response:
Thank you for your suggestion. Due to the material directly growing on the carbon cloth substrate, conducting BET and IR experiments would not yield meaningful results. As we have already established the phase via XRD and analyzed the elemental composition using XPS, we believe EDX analysis is not necessary.
Comment4:
The authors should do the analysis the conclusion section must clearly establish a strong correlation with the proposed topic.
Authors’ response:
Thank you for your comment. We have revised the title and conclusion to more clearly align with the proposed topic as follows:
Title: Electrochemical detection of ammonia in water using NiCu carbonate hydroxide modified carbon cloth electrodes: a simple sensing method
Conclusion: In summary, we have successfully synthesized a novel self-supporting electrode via hydrothermal methods, producing nickel-copper carbonate hydroxide material on a carbon cloth current collector. This electrode demonstrates rapid and highly sensitive detection of ammonia nitrogen in water, characterized by excellent catalytic activity, reasonable charge transfer properties, and good stability. It exhibits high sensitivity with 3.9 μA μM⁻¹ cm⁻² in the range of 1 μM to 100 μM and 3.13 μA μM⁻¹ cm⁻² in the range of 100 μM to 400 μM, with a detection limit as low as 0.62 μM. While its sensitivity does not match that of precious metal-based materials, it offers practical performance at a significantly lower cost. Furthermore, tested in tap water, it demonstrated good recovery rates, underscoring its potential for real-world applications. Its stability and sensitivity in the presence of common interfering ions suggest its suitability for practical water quality assessment.
These revisions aim to establish a clearer correlation with the study's objectives and outcomes.
Comment5:
Single and plurality, upper and lower scripts, punctuation and other details should be checked carefully
Authors’ response:
Thank you for your comment. We have carefully reviewed the manuscript to ensure consistency in singular and plural terms, upper and lower scripts, punctuation, and other details.
Comment6:
The previous work on this topic may be updated in the introduction, Chemosphere, 2022, 307,135729; Sol. RRL, 2023,7, 2300143; Molecules 2023, 28, 4507.
Authors’ response:
Thank you for your comment. We have updated the introduction to include a reference to the article in Chemosphere, 2022, 307, 135729, which reviews various applications of Ni-MOFs in the electrochemical field relevant to our work. However, we have not cited the other two articles as they describe an organic heterojunction solar cell and a polymer for electrolysis, respectively, which have weaker relevance to our study.
Comment7:
I suggest the authors have to explain the mechanism in detail.
Authors’ response:
Thank you for your suggestion. We have preliminarily discussed the possible mechanisms in sections 3.2 and 3.3. For ammonia catalysis, there are typically two pathways, which share the same initial stage. However, there is significant debate regarding these pathways, making it challenging to determine the exact mechanism. A detailed discussion would require in-situ Raman spectroscopy and other advanced techniques, which are currently beyond our capabilities.

Round 2
Reviewer 1 Report
Comments and Suggestions for Authors
The manuscript has been significantly improved upon the addressing of reviewer comments, I can now recommend publication.
Author Response
Thank you for your feedback and recommendation for publication. We appreciate your time and effort in reviewing our manuscript.
Reviewer 2 Report
Comments and Suggestions for Authors
1. Recovery experiments at LOQ levels should be included.
2. There are still many formatting issues. For example, table 1, 8.69μM → 8.69 μM, Pt-Ni(OH)2 → Pt-Ni(OH)2, 1000 → 1,000; Line 327, Table 2..; table 2, Found(μM) → Found (μM), %,n=3 → %, n=3; line 39, electrode [12; line 153, [36] appear
You should carefully review the entire manuscript, including the text in the figures, rather than focusing solely on the mentioned issues.
Author Response
Comment 1:
Recovery experiments at LOQ levels should be included.
Authors’ response:
Thank you for your valuable suggestion. Based on our calculations, the limit of quantification (LOQ) is approximately 2 μM (S/N=10). However, after several attempts, we found it challenging to obtain accurate data at this concentration. One possible reason is the slower charge transfer of hydroxides and the multilayer structure during catalysis, which causes slower response at low concentrations and results in relatively unstable baseline, leading to higher limits of detection (LOD) and quantification (LOQ). While this issue is not prominent at higher concentrations, it becomes critical at lower concentrations. We acknowledge that further improvements to this material are needed.
Future work will focus on controlling morphology through adjustments in concentration, hydrothermal temperature, and time, and improving conductivity by intercalating surfactants. Additionally, designing specialized circuits for the sensor's frontend electrode may enhance its performance. Despite these challenges, we believe the electrode has significant value due to its cost-effectiveness compared to precious metal electrodes.
Comment 2:
There are still many formatting issues. For example, table 1, 8.69μM → 8.69 μM, Pt-Ni(OH)2 → Pt-Ni(OH)2, 1000 → 1,000; Line 327, Table 2..; table 2, Found(μM) → Found (μM), %,n=3 → %, n=3; line 39, electrode [12; line 153, [36] appear.
Authors’ response:
Thank you for pointing out the formatting issues. We have carefully reviewed the entire manuscript, including the text in the figures, and have corrected all formatting issues.
